# Detection of Lassa Virus-Reactive IgG Antibodies in Wild Rodents: Validation of a Capture Enzyme-Linked Immunological Assay

**DOI:** 10.3390/v14050993

**Published:** 2022-05-07

**Authors:** Hugo Soubrier, Umaru Bangura, Chris Hoffmann, Ayodeji Olayemi, Adetunji Samuel Adesina, Stephan Günther, Lisa Oestereich, Elisabeth Fichet-Calvet

**Affiliations:** 1Bernhard Nocht Institute for Tropical Medicine, 20359 Hamburg, Germany; umaru.bangura@bnitm.de (U.B.); hoffmann@bnitm.de (C.H.); guenther@bni.uni-hamburg.de (S.G.); oestereich@bni-hamburg.de (L.O.); 2Natural History Museum, Obafemi Awolowo University, Ile Ife HO220005, Nigeria; ayyolayemi@yahoo.com; 3Department of Biochemistry and Molecular Biology, Obafemi Awolowo University, Ile Ife HO220005, Nigeria; asadesina@yahoo.com; 4German Center for Infectious Diseases (DZIF), Partner Site Hamburg-Lübeck-Borstel-Riems, 20359 Hamburg, Germany

**Keywords:** ELISA, IFA, *Mastomys*, natural host, IgG, Lassa virus, West Africa

## Abstract

The aim of this study was to evaluate the use of a capture enzyme-linked immunosorbent assay (ELISA) for the detection of LASV-reactive IgG antibodies in *Mastomys* rodents. The assay was used for laboratory-bred *Mastomys* rodents, as well as for animals caught in the wild in various regions of West Africa. The ELISA reached an accuracy of 97.1% in samples of known exposure, and a comparison to an immunofluorescence assay (IFA) revealed a very strong agreement between the ELISA and IFA results (Cohen’s kappa of 0.81). The agreement is valid in Nigeria, and in Guinea and Sierra Leone where the lineages II and IV are circulating, respectively. Altogether, these results indicate that this capture ELISA is suitable for LASV IgG serostatus determination in *Mastomys* rodents as an alternative to IFA. This assay will be a strong, accurate, and semi-quantitative alternative for rodent seroprevalence studies that does not depend on biosafety level 4 infrastructures, providing great benefits for ecology and epidemiology studies of Lassa fever, a disease listed on the Research and Development Blueprint of the WHO.

## 1. Introduction

Lassa fever (LF) is a haemorrhagic fever endemic in West Africa caused by the Lassa virus (LASV), an RNA virus of the *Arenaviridae* family [1]. The main reservoir is the natal multimammate mouse *(Mastomys natalensis)*, but recent study has also indicated that other rodent species (including *Mastomys erythroleucus* and *Hylomyscus pamfi*) could also be hosts and likely participate in the virus life cycle [2,3]. Transmission to humans occurs either through indirect contact (the consumption of food contaminated by infected rodent excreta, or the inhalation of aerosolized particles), or by direct contact with infected rodents (contact or consumption). Further human to human transmission can occur, and lead to epidemics, with the nosocomial context being of particular importance [4]. Seroprevalence studies across endemic countries have revealed high IgG seropositivity rates that do not match the observed prevalence of human disease, suggesting that most cases are actually pauci- or asymptomatic [5,6]. Additionally, reservoir seroprevalence studies are of crucial importance to understand the ecology, life cycle and overall epidemiology of LASV. Seroprevalence studies also provide a base to understand the seasonality of LASV infection and its dynamics in the rodent reservoir population [2,7].

The classical approach to detect IgG and IgM antibodies against LASV has been the immunofluorescent assay (IFA), using virus-infected cells as antigens. Described by Wulff and Lange in 1975 [8], this technique has been used as the basis of many seroprevalence studies, and is actually the basis for the main cited LF incidence study [9,10]. However, it is dependent on biosafety level 4 (BSL-4) infrastructures for the manipulation of infected cells, and also heavily relies on skilled investigators to interpret fluorescence signals. Enzyme-linked immunosorbent assays (ELISAs) provide an alternative for the detection of anti-Lassa IgG and IgM. Independent of BSL-4 facilities, straightforward and semi-quantitative, ELISAs are very convenient for seroprevalence studies, especially in regions with limited resources. Bausch et al. [11] indicated that the sensitivity and specificity rates were higher than with IFA and, nowadays, the technique is extensively used in human seroprevalence studies [12,13,14]. Despite the use of ELISAs with human samples, rodent seroprevalence studies remain largely dependent on classical IFA [15,16,17,18]. To our knowledge, only Demby et al. used both an ELISA and IFA to investigate LASV antigens and the IgG of rodents in Guinea. Using the ELISA protocol described for humans previously, they showed that ELISA was safe, rapid and a more affordable technique for this kind of study [2,11].

The aim of our study was to investigate the use of a capture ELISA described in Gabriel et al. [13] for the detection of anti-LASV IgG in the rodent populations of West Africa. Using rodent samples collected in various sites across Guinea, Sierra Leone, and Nigeria, as well as samples from laboratory-bred *Mastomys*, we compared seroprevalences using both the IFA and the capture-ELISA. Additionally, we investigated the impact of dry blood spot storage and its subsequent elution on the ELISA results, compared to whole blood storage.

## 2. Materials and Methods

### 2.1. Origin of Laboratory Samples

Blood samples from the laboratory-raised *M. natalensis* were used to evaluate the ELISA’s specificity and sensitivity. The breeding colony was established thanks to a collaboration with the Rocky Mountain laboratory [19]. The animals descended from wild-caught, arenavirus-free animals from Mali, and are now bred in our animal facility at the Bernhard Nocht Institute for Tropical Medicine (BNITM). Of 104 animals, 62 (aged between 6 and 59 days) were subcutaneously inoculated with 1000 focus-forming units (FFU) of LASV Bantou 366 strain (GenBank n° GU830839, *n* = 45) and Kako 428 strain (GenBank n° KT992425, *n* = 17) in 50–100 uL PBS. In order to include animals with a full immune reaction with the synthesis of immunoglobulins G, we used the blood from animals sampled between 11 and 106 days post-inoculation (d.p.i). This was based on another experiment conducted on *Mastomys* infected with Morogoro virus, which showed that all animals were seropositive after 7 days, without the antibodies disappearing over time [16]. Depending on the age and weight of the animals, 100−200 µL of blood was drawn by puncturing the tail or the saphenous vein once per week. Larger quantities (up to 1 mL) were obtained via cardiac puncture following euthanasia. Animals were euthanized via isoflurane overdose followed by decapitation. These animals were used as a positive cohort. Blood samples from 42 naïve animals were used as a negative cohort. All the rodents were tested for the presence of anti-LASV IgG with the ELISA and IFA methods. All animal procedures were carried out in strict compliance with the recommendations of the German Society for Laboratory Animal Science under the supervision of a veterinarian. All protocols were approved by the Committee on the Ethics of Animal Experiments of the City of Hamburg (Permit No. 32/14, N 028/2018 and N 050/2021).

### 2.2. Origin of Wild Samples

The rodent blood samples included were part of a large collection from previous studies conducted in West Africa. Blood samples from a total of 361 wild *M. natalensis* (*n* = 297) and *M. erythroleucus* (*n* = 64) were tested by both ELISA and IFA. They were categorized depending on their geographic origin and the LASV lineages circulating in their area of origin.

In Guinea, 118 rodents were captured in 3 villages (Brissa, Madina Oula and Yarawalia) where lineage IV of LASV circulates. *M. erythroleucus* accounted for 9 of the samples from Madina Oula [3,17] (Figure 1).

In Sierra Leone, 106 animals were captured in 6 villages (Baoma, Falaba, Makakura, Ngolahun, Nyandeyama, and Sambehun), where LASV lineage IV also circulates [15]. *M. erythroleucus* accounted for 1 sample from Makakura and Sambehun each (Figure 1).

In Nigeria, 137 rodents were captured in 5 villages (Eguare-Egoro, Ekpoma, Mayo Ranewo, Ngel Nyaki, and Onmba-abena), where LASV lineages II and III circulate. *M. erythroleucus* accounted for 53 samples from Onmba-abena [3] (Figure 1). The samples were collected according to local legislation, and the trapping was approved by the Ethics Committee of the Ministry of Health and Sanitation, Government of Sierra Leone (16 May 2012) and the National Ethic Committee, Guinea (12/CNERS/12). Ethical review and approval were waived for the study conducted in Nigeria, due to the absence of human sampling.

### 2.3. ELISA

The ELISA was performed using the BLACKBOX^®^ LASV IgG ELISA Kit developed by the Diagnostics Development Laboratory hosted at BNITM. The assay was based on Gabriel et al., 2018. Prior to handling in the BSL-2 lab, all blood samples were inactivated by adding Triton-X 100 to a final concentration of 1%.

Then, 1 µL of the inactivated whole blood sample was mixed with the provided dilution buffer (1:50) and incubated with a biotinylated recombinant LASV nucleoprotein (NP) antigen (LASV AV strain, GenBank n°: AF246121). The antigen was recombinantly expressed in bacteria (*Escherichia coli*), which differed slightly from the initial protocol which used the insect cells (*Spodoptera frugiperda*). ELISA plates were coated with a recombinant human CD32, which constituted the IgG immune complex-specific capture molecule (see details in [13,20]). Three negative controls and one positive control provided by the kit were added. The plates were left for 24 h incubation at 4–8 °C in a wet chamber. The following day, plates were thoroughly washed using a wash buffer, and 1:10,000 diluted HRP-labelled streptavidin was added to all wells and incubated at 4–8 °C in a wet chamber for 1 hr. A final wash was performed before the addition of 100 µL of 3,3’,5,5’-Tetramethylbenzidine (TMB) substrate to all the wells, which were then incubated for 10 min. The colorimetric reaction was stopped by adding 100 µL of STOP solution (sulfuric acid).

The optical density at 450 nm and 620 nm was measured using a microplate reader within 30 min of stopping the assay and results were interpreted using the optical density difference between 450 and 620 for each well. The cut-off value was defined as the mean values of the three negative controls + 0.150, as stated in the kit manual. Secondly, for each well, the Index Value (IV) was calculated as the OD difference divided by the cut-off value. Results were considered IgG seropositive with index values greater or equal to 1.1; negative results had index values less or equal to 0.9, and the results were ambiguous when the index value lay between 0.9 and 1.1.

### 2.4. IFA

Prior to handling on a BSL-2 bench, all samples were 1:1 inactivated by adding Triton-X 100 at a final concentration of 1%. The method used to perform IFA testing was based on the original work of Wulff and Lange, 1975 [8]. Briefly, LASV (Bantou strain) was propagated on Vero cells in the BSL 4 laboratory at BNITM. Cells were acetone-fixed on immunofluorescent slides. Undiluted inactivated whole-blood samples were added to the slides and incubated for 1h at 37 °C. Slides were then washed with phosphate-buffered saline (PBS). Bound antibodies were detected by adding Fluorescein isothiocyanate (FITC)-labelled anti-mouse immunoglobulin G at a dilution of 1:200, and incubated for 1hr at 37 °C. After a final wash, slides were observed under fluorescent microscopy. Cells that showed specific anti-LASV staining under the microscope were considered positive. The analysis of the slides was performed by two independent persons.

### 2.5. Elution of Dried Blood on Filter Paper

A matched subset of whole blood samples (*n* = 108) was also tested using filter paper eluates from dried blood spots. These samples were eluted in a 300 µL solution of 1× PBS containing 0.2% concentration of liquid ammonium and 2.5 µL of a 25% NH_3_ solution [21]. The eluates were then used without further dilutions for ELISA. Both whole blood and dried blood spots were collected in two villages in Guinea (Brissa and Yarawalia).

### 2.6. Statistical Analysis

Samples from laboratory-bred-infected (positive) and -uninfected (negative) cohorts were used to estimate the specificity, sensitivity and accuracy of the ELISA.

**Specificity** is defined as the proportion of true negatives that the test can accurately detect. It is calculated as:$$Specificity=\frac{N\;negative\;ELISA\;in\;negative\;cohort}{N\;negative\;cohort}$$

**Sensitivity** is defined as the proportion of true positives that the test can accurately detect, and it is calculated as:$$Sensitivity=\frac{N\;positive\;ELISA\;in\;positive\;cohort}{N\;positive\;cohort}$$

**Accuracy** is defined as the proportion of samples that are correctly classified by the ELISA. It is calculated as:$$Accuracy=\frac{\left(N\;positive\;ELISA\;in\;positive\;cohort+N\;negative\;ELISA\;in\;negative\;cohort\right)}{N\;all\;tested}$$

**Inter-rater reliability:** in order to evaluate the agreement between the ELISA and IFA, two kinds of measures were used.

Firstly, the percent agreement was used as an initial evaluation of the agreement between the tests. The overall percent agreement is calculated as follows:$$Po=\frac{\left(N\;positive\;both\;ELISA\;and\;IFA+N\;negative\;both\;ELISA\;and\;IFA\right)}{N\;all\;tested}\;$$

The overall agreement is useful, but may be biased as it does not differentiate between the agreement within the negative and positive results. Excessive amounts of agreement within positive or negative results would yield high overall agreement, even if the other group showed a much lower agreement.

Therefore, negative and positive agreements were also compared to evaluate the agreement specifically within the negative and positives results. This calculation closely resembles the specificity and sensitivity parameters. However, here, comparison is performed against the IFA, which does not always produce correct classification, and would therefore bias the sensitivity and specificity estimates. Therefore, this is a question of agreement between the assays rather than specificity and sensitivity [22,23].
$$Positive\;Agreement=\frac{N\;positive\;both\;IFA\;and\;ELISA}{N\;IFA\;positive}\;$$
$$Negative\;Agreement=\frac{N\;negative\;both\;IFA\;and\;ELISA}{N\;IFA\;negative}\;$$

Additionally, for a more robust evaluation of the concordance between assays, the degree of agreement was determined using Cohen’s kappa. This has the advantage of accounting for agreements that arise due to chance.
$$k=\frac{observed\;agreement-expected\;agreement\;}{1-expected\;agreement}$$

A Kappa value of 1 indicates perfect agreement, while a value of 0 indicates no agreement better than chance [24].

Differences in proportions of positive results between ELISA and IFA were investigated using the Z-test. Mean index values were compared using two-way Student’s *t*-tests. The McNemar test, a modification of the ordinary Chi-square test that takes into account the paired nature of response data, was also used to investigate systematic difference between the assays. All statistical analysis and data visualization were performed using the R software [25].

## 3. Results

### 3.1. Laboratory-Bred Mastomys

All the animals (*n* = 42) in the negative cohort were tested negative by ELISA, leading to a specificity of 100%. Of the 62 animals belonging to the positive cohort, 59 rodents tested positive by the ELISA, leading to a sensitivity of 95%. The accuracy of the ELISA was therefore 97.1% (93.8–100%) for laboratory-raised *Mastomys*. (Table 1). All the animals of the negative cohort tested negative by the IFA, and 60 animals of the positive cohort tested positive by the IFA.

We did not observe any difference in sensitivity between the two LASV strains. Amongst the three ELISA negatives, two were inoculated with the LASV Bantou 366 strain and one was inoculated with the LASV Kako 428 strain.

### 3.2. Wild Mastomys

Of the 361 samples of wild rodents, 85 (23.5%) tested positive by the ELISA, and 90 (24.9%) tested positive by the IFA. This difference in proportions between the ELISA and IFA positives was not statistically significant (Z-test *p*-values all > 0.7, Table 2). In addition, there was no evidence of a statistical difference in ELISA- or IFA-positive proportions between the three countries (Z-test *p*-values > 0.08). The seroprevalence by recorded by the ELISA and IFA was therefore similar, in the ranges of 18.9–28.8% and 20.8–29.7%, respectively (Table 2).

Figure 2 indicates that some misclassification does occur, and some IFA-negative samples exhibited an ELISA index value over 1.1. Similarly, there were some IFA-positive samples that exhibited an ELISA index value below 1.1. However, overall, samples that were IFA-negative had a mean index value of 0.37 (95% CI: −0.90–1.64) in the ELISA (Figure 2, Appendix A). Samples that were IFA-positive had a mean index value of 3.67 (95% CI: −1.65–8.99) in the ELISA. The Student’s t-test indicated that there was very strong evidence for a difference between the negative and positive outcomes (*p*-values < 1.8 × 10^−8^).

### 3.3. Concordance between the Tests

Using the percent agreement, both assays agreed on the classification of more than 93% of all samples (93.1%, 95% CI: 90.5–95.7). This was confirmed by a high Cohen’s kappa of 0.811 (95% CI: 0.74–0.883, SD: 0.036). Looking at individual countries, we observed the lowest agreement in Nigeria (91.2%, 95% CI: 86.5–96). This was supported by the lowest Cohen’s kappa in Nigeria (0.755, 95% CI: 0.623–0.888) and the highest Cohen’s kappa in Guinea (0.857, 95% CI: 0.754–0.96). These estimates had overlapping confidence intervals, suggesting no statistical differences between them. Additionally, McNemar tests indicated no evidence of systematic bias between the assay results, suggesting symmetry in the contingency tables (*p*-values > 0.05 for countries and full dataset) (Table 3 and Table 4).

However, some discrepancies could not be resolved: of the 361 samples, 25 (6.9%) were misclassified between the two assays. N = 26 ELISA-negative samples were classified as positive by IFA, and *n* = 10 ELISA-positive samples were classified as negative by IFA (Table 3). This represented 9% of all ELISA-negative samples and 12% of all ELISA-positive samples. Seven of these samples were from Nigeria, while Guinea and Sierra Leone accounted for four samples each. Interestingly, two of these discordant samples from Nigeria also tested positive when tested by IFA for other arenaviruses (the Morogoro and Mobala viruses).

### 3.4. Whole Blood vs. Dried Blood on Filter Paper

When eluting dried blood from filter papers, we qualitatively observed various degrees of blood elution quality, using the color of eluates after overnight elution. While most samples could be easily eluted, *n* = 14 samples showed poor elution, and *n* = 8 samples could not be eluted at all (transparent elution). We therefore removed these last eight from the analyses.

The ELISA using whole blood classified 29 out of 100 dried blood spot samples as positive (29%). The ELISA using the same eluates classified 15 samples as positive (15%), and 4 samples as ambiguous. A Z-test for the comparison of proportions indicated that there was a difference between the proportions of positive samples when using the whole blood against filter paper eluates (*p*-value: 0.026, X^2^ = 4.92) (Figure 3, Appendix A).

The mean ELISA index value of the positive whole blood samples was 5.645 (SD: 3.135, 95% CI: 0.509–11.798) compared with a mean index value of 1.228 (SD: 0.935, 95% CI: 0.605–3.061) for the filter paper eluates of those same samples. The difference was statistically significant (Student *t*-test t: 7.26, *p*-value: 2.5 × 10^−8^) with a 4.6-fold decrease in the ELISA index value following elution (Figure 3). Samples with weak elution showed low index values of 0.426 (SD: 0.684, 95% CI: −0.914–1.766), classifying them as negative. However, inside this panel, some samples were detected as positive with whole blood, showing a mean index value of 8.15 (SD: 2.386, 95% CI: 3.476–12.829). In addition, the four samples that were classified as ambiguous with filter paper eluates had high index values (<3) with whole blood.

## 4. Discussion

This study showed that the use of a capture ELISA is a good alternative to the use of IFA for the detection of anti-LASV IgG in *Mastomys* spp. In laboratory-bred animals with a known and controlled exposure to LASV, the test performed very well and the assay accuracy of 97.1% (93.8–100), the specificity of 100%, and the sensitivity of 95% were similar to those obtained in humans (91.5–94.3%, 100% and 90%, respectively, in Gabriel et al., 2018) [13]. Although no test is perfect, we could try to increase the sensitivity by testing negative samples twice with different dilutions. Indeed, the negative ELISA result but positive IFA in the positive cohort may be due to a low amount of antibodies against NP, whereas a high amount of antibodies against glycoprotein (GP) led to a positive IFA. The sensitivity remained similar in rodents infected with two distant strains of LASV: Bantou 366 (lineage IV) and Kako 428 (lineage VI). This suggests that this ELISA kit, based on the AV strain belonging to lineage V, is capable of detecting antibodies produced by other strains belonging to lineages IV and VI.

In wild animals coming from Guinea, Sierra Leone and Nigeria that are infected by different LASV lineages, the assay also performed well. Importantly, the ELISA showed good specificity and sensitivity using frozen whole blood from wild rodents, which is not used in routine serology, as most experiments are completed with plasma or serum. There was no indication that the presence of cells in the whole blood may have impacted the validity of our results. This is of interest for field studies, especially on rodents, for which serum or plasma can be difficult to obtain, either due to the low resource settings or the small blood volumes due to the small body size of the animals. Additionally, our results remained valid when investigating *M. erythroleucus* instead of *M. natalensis* samples.

### 4.1. ELISA versus IFA

It was observed that the IFA and ELISA produced a very similar serostatus classification in *Mastomys* spp. They showed strong agreement, reaching a Cohen’s kappa of 0.811 for the global dataset. The agreement between the tests remained high when looking at countries with different LASV lineages circulating.

Interestingly, 9% of ELISA’s negative results had positive results in IFA. As the sensitivity of this ELISA was 95% in laboratory-infected *Mastomys*, this was higher than the expected 5% of false negatives. Arguably, we can consider that animals which were positive by IFA were either false IFA-positive, due to IFA subjective interpretation, or true IFA-positive for other arenaviruses. Indeed, IFA is known for broad reactivity, so cross-reactivity between arenaviruses is highly possible [9,26,27,28,29]. The well-described co-circulation of other closely related arenaviruses, such as Mobala virus, in *Mastomys* populations in Mayo-Raneyo in Nigeria [18] is highly likely to create false positive results due to cross-reactivity, and the previous exposure of wild rodents to such arenaviruses in Nigeria could explain the lower agreement in this country. Samples from Nigeria showed the greatest difference between the IFA and ELISA seropositive results, with the lowest agreement between the assays, as well as the greatest difference between the positive and negative agreements. In this situation, we argue that the ELISA is more specific and could provide a more reliable determination of the true serostatus of anti-LASV IgG in *Mastomys*. Notably, Bausch et al. (2000) have already pointed towards a higher sensitivity and specificity of the ELISA compared to the IFA in humans [11]. This is a recurrent aspect of ELISA and IFA comparisons, even in contexts outside Lassa fever or other infectious diseases, such as schistosomiasis, leishmaniasis, or hantavirus disease [30,31,32].

Additionally, it was observed that 12% of ELISA-positive samples from wild animals were negative in the IFA. These may be IFA false negatives, but it is also possible that the relatively low sample size for the laboratory cohorts may have overestimated the specificity of the ELISA. Moreover, it could be that these animals only had a mild or recent exposure to LASV, and therefore did not have a lot of IgG. The individual rodents in question may have had a low titer of anti-NP IgG to be detected by ELISA, but not enough to produce a positive fluorescence signal by IFA, especially taking into account the subjectivity of IFA interpretation.

Diagnostic tests are never perfect, and will always result in bias towards a result. It is important to measure and acknowledge this bias, and the balance between false negatives and false positives must be carefully taken into account. In the context of LF, we consider that the capacity to detect positives rates prevails on the capacity to detect negatives rates, which would lead to seropositivity underestimation. Despite this, misclassification remained marginal throughout our panel of tests, representing less than 7% of the samples.

### 4.2. Dried Blood Spot and Serology

While the use of dried blood spots on filter paper can be very useful for on-ground collection, transport, and storage, it appears that it is less suitable for the ELISA. Indeed, a 4.6-fold decrease in ELISA index values indicated that the antibody levels within the filter paper eluates were greatly reduced. As a result, many samples that are seropositive using whole blood could end up being classified as negative or ambiguous. This decrease in antibody levels is either caused by the degradation of antibodies within the filter paper, trapping of the IgG molecules inside the blood clot and the fibers of the paper, bad drying of the paper on the field (i.e., under the sun), a poor elution process, or the over-dilution of samples when eluting. Internal tests showed that 300 µL buffer is the minimum requirement for elution, as lower volumes do not allow for efficient mixing, and thus result in even poorer elution.

Overall, the use of eluates instead of whole blood reduced the seroprevalence in our test panel by half, from 29% to 15%. These results confirm the comparison we had already completed with samples collected in Sierra Leone and tested by IFA, where a loss of 20% in IgG prevalence was observed when using the dried blood spot [15]. Given the common use of dried blood spots in arenavirus rodent seroprevalence studies [33,34,35], extra care must be taken when interpreting results by ELISA or IFA, as seroprevalence could be underestimated. Studies that quantitatively investigate antibody status can account for the loss of antibodies due to the elution process by normalizing the results on a standard curve of the spectrophotometric absorption after elution in PBS, as detailed in Borremans (2014) [21].

Conclusively, our study indicates that this capture ELISA is highly suitable for the detection of anti-LASV IgG in rodents, and complementary to the testing performed in humans. As a much more straightforward process that does not require BSL 4 practices, this ELISA is very suitable for the direct serostatus determination of *Mastomys* in endemic countries that lack such biosafety infrastructures. Importantly, ELISA result analysis is quantitative, and can be automated, unlike IFA analysis where classification is highly subjective and requires blinded and particularly skilled examinators.

## Figures and Tables

**Figure 1 viruses-14-00993-f001:**
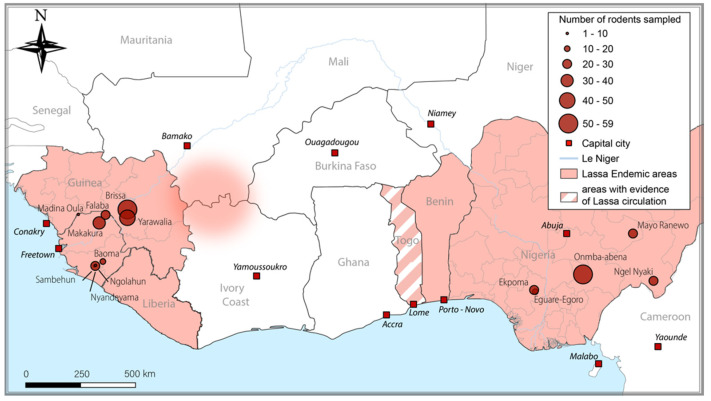
Map of Lassa endemic regions and study sites across West Africa. The coordinates are provided in Appendix A.

**Figure 2 viruses-14-00993-f002:**
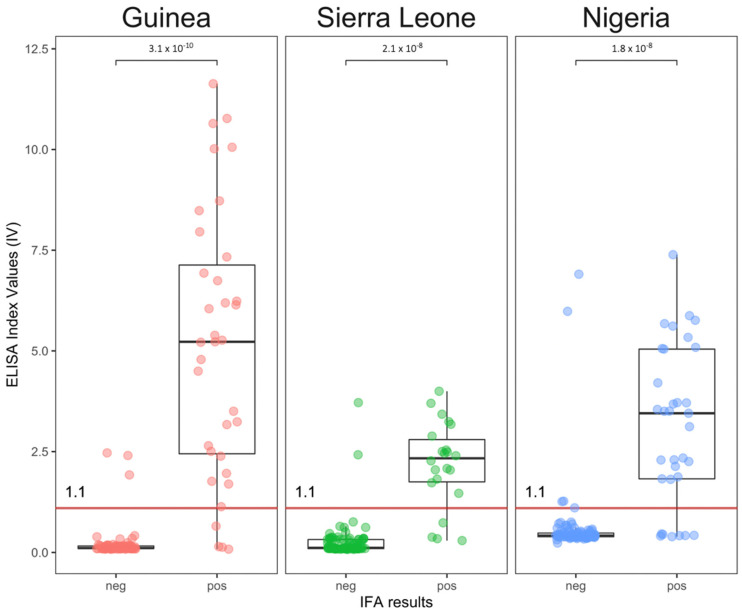
Distribution of ELISA index values by IFA outcome and by country of sampling.

**Figure 3 viruses-14-00993-f003:**
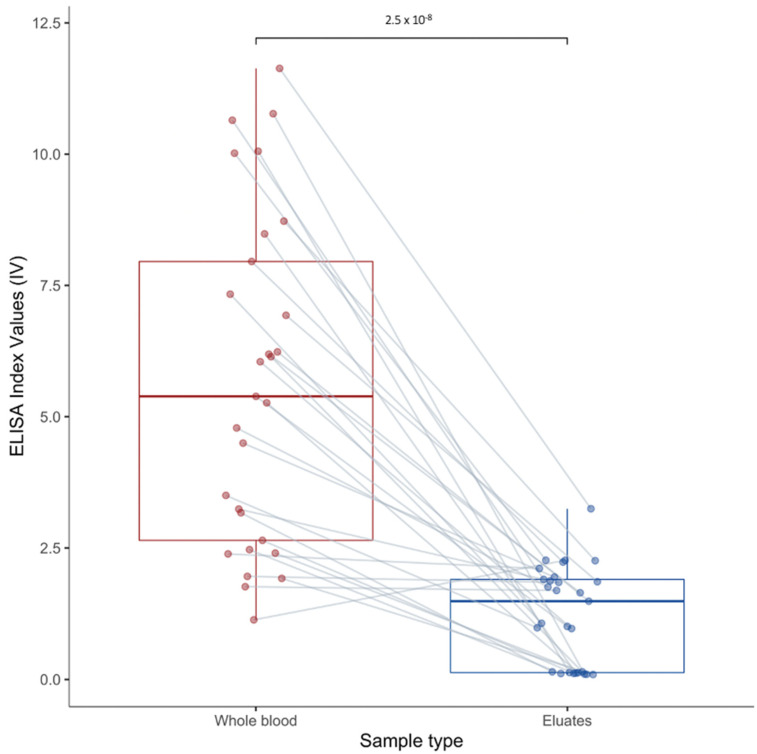
Mean ELISA index value of whole blood-positive samples dispatched by type of biopsy: full blood (in red) versus dried blood on filter paper (in blue).

**Table 1 viruses-14-00993-t001:** ELISA results for the laboratory-bred cohort of positive animals (inoculated), and negative animals (not inoculated).

Cohort	ELISA +	ELISA −	Total
Positive	59	3	62
Negative	0	42	42
Total	59	45	104

**Table 2 viruses-14-00993-t002:** LASV seroprevalence by ELISA and by IFA results per country.

Country	N	ELISA + (% of N)	IFA + (% of N)	*p*-Values *	X^2^
Guinea	118	34 (28.8)	35 (29.7)	0.886	0.021
Sierra Leone	106	20 (18.9)	22 (20.8)	0.730	0.119
Nigeria	137	31 (22.6)	33 (24.1)	0.775	0.082
Total	361	85 (23.5)	90 (24.9)	0.664	0.189

* *p*-values from Z-test for comparing proportions of ELISA-positive and IFA-positive results.

**Table 3 viruses-14-00993-t003:** Contingency table of IFA and ELISA results for the full dataset with row percentages.

	IFA − (%)	IFA + (%)	Total
**ELISA −**	250 (91)	26 (9)	**276**
**ELISA +**	10 (12)	75 (88)	**85**
**Total**	**260 (75)**	**101 (25)**	**361**

**Table 4 viruses-14-00993-t004:** Summary of agreement measures between assays by geographic regions.

	Guinea (95% CI)	Sierra Leone (95% CI)	Nigeria (95% CI)	Full Dataset (95% CI)
Overall agreement	94.1% (89.8–98.3)	94.3% (89.9–98.7)	91.2% (86.5–96)	93.1% (90.5–95.7)
Positive agreement	88.6% (78–99.1)	81.8% (65.7–97.9)	78.8% (64.8–92.7)	83.3% (75.6–91)
Negative agreement	96.4% (92.4–100.4)	97.6% (94.4–100.9)	95.2% (91.1–99.3)	96.3% (94.1–98.6)
Cohen kappa	0.857 (0.75–0.96)	0.822 (0.68–0.96)	0.755 (0.62–0.89)	0.811 (0.74–0.88)

## Data Availability

The data presented in this study are available in the Appendix A.

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
