# Peer review of "Detection of Lassa Virus-Reactive IgG Antibodies in Wild Rodents: Validation of a Capture Enzyme-Linked Immunological Assay"

_viruses, 2022, doi:10.3390/v14050993_

Round 1
Reviewer 1 Report
In this manuscript Soubreier et al evaluate the use of a capture ELISA for detection of LASV IgG in Mastomys rodents. Using rodent samples collected in various sites across Guinea, Sierra Leone and Nigeria, along with lab infected Mastomys samples, the authors compared seroprevalences using both the IFA and the capture-ELISA. Additionally, They also investigated the impact of dry blood spot storage and subsequent elution on ELISA results compared to whole blood storage. . The paper addresses an important problem of LASV surveillance in resource-poor setting. The ELISA assay improves the throughput while also reducing sampling processing times.
Overall, this was an interesting paper and a well-organized manuscript. I have no serious concerns with the methodologies or any major issues with the manuscript in general.
However, a few minor concerns need to be addressed:
- For the lab samples, can the authors include information about when the positive samples were taken? How many days post infection were the samples taken? What volume of samples were drawn?
- On line 53: It should read “Independent of BSL-4 facilities” instead of “Independent on BSL-4 facilities”
Author Response
We thank the reviewer 1 for his/her positive comments. Below are the responses to the minor comments.
1- We included information about blood sampling in captive and inoculated Mastomys in the section 2.1. The blood of the animals was sampled at different days post infection from day 11 to day 106, and the volume was 100-200µl during the follow-up study. This information is now detailed in lines 80-88.
Because of these new details, we also added 2 animals which were forgotten in the previous version. The total number is now 104 (instead of 102), including 62 in the positive cohort, and 42 in the negative cohort. The table 1 changed accordingly.
2- correction done
Reviewer 2 Report
In this manuscript, authors evaluated the use of a capture enzyme-linked immunosorbent assay for detection of LASV IgG in Mastomys rodents. Additionally, they also investigated the impact of dry blood spot storage and subsequent elution on ELISA results compared to whole blood. This paper containing interesting results and discussion, however I still have some suggestions below.
- Materials and Methods Section 2.1: Authors should clearly state how soon after virus inoculation blood is drawn. Since IgM and IgG are produced at different times, the rationale for the timing of blood collection should also be explained.
- Line 314: Here, authors argued that the ELISA is more specific as it relies on the capture of a recombinant Lassa virus specific Nucleoprotein antigen, and could provide more reliable determination of true serostatus of anti-Lassa IgG in Mastomys. There may be different opinions here. Since the viral nucleoprotein is located within the viral envelope, the production of antibodies against it may not be much greater than that of anti-envelope protein antibodies. So these IFA-positive, but ELISA-negative whole blood samples, how to rule out that it is not caused by too few anti-nucleoprotein antibodies within whole blood samples?
- It would be better if there are other arenavirus antibodies to test specificity or recombinant LASV GP2 protein as an ELISA antigen for comparison.
Author Response
We thank the reviewer 2 for his/her valuable comments and make our detailed response below.
1- An experiment conducted in Mastomys infected with Morogoro virus showed that all animals were seropositive after 7 days, without the antibodies disappearing over time (Borremans et al. 2015). Thus, we chose to include data from rodents tested between 11 and 106 days. This information is now included in the section 2.1 (lines 80-84).
2- The reviewer is right; this is not because the capture ELISA is built with the nucleoprotein that it is more specific in the context of testing the Nigerian samples. This was because the samples from Mayo-Raneyo were infected by a Mobala-like virus, and not by LASV. We therefore removed this sentence from this part of the discussion.
However, your comment about a low quantity of antibodies against NP, whereas a higher quantity of antibodies against GP could lead to a negative ELISA test and a positive IFA test is interesting. We could solve this discrepancy by testing twice each negative sample with 2 different dilutions. This is now included in the discussion (line 320-323).
3- Nucleoprotein was chosen as antigen, as data from Lassa fever survivors indicate that most of the B cell response is directed against this protein and GP antibodies arise only late. In addition, nucleoprotein is easy to express in bacteria. We agree that looking for other antibody specificities may add to the study. However, production of recombinant glycoproteins is quite challenging requiring eucaryotic or even mammalian expression systems. Setting up such as system and generation of evaluation data may well take several months and may not be accomplished within the time frame of this revision. Unfortunately, we cannot use commercial assays. The ELISA used in the study is based on our proprietary Fc-gamma receptor technology. This receptor is not so species-specific, and therefore, this assay is suitable to detect rodent antibodies.
Reviewer 3 Report
This manuscript reports the use of an ELISA for the detection of LASV IgG in rodents. The ELISA was previously developed and tested on human samples with success. Results presented in this study indicate that this assay is also suitable for rodent samples and may therefore offer a more practical assay than IFA for the detection of LASV IgG, particularly on the field. This ELISA would be an asset in west African countries for LASV surveillance and seroprevalence studies in the rodent population.
Authors validated their assay in experimentally infected rodents and on field samples and in both cases, the test performed very well compared to IFA. They also evaluated the performance of their assay on dried blood eluted from filter paper and results indicated that such samples would reduce the sensitivity of their assay, suggesting that such on-ground habit should be avoided to obtain reliable results.
The ELISA performed well on Guinean samples but less on Sierra Leonean and Nigerian samples. The low agreement between ELISA and IFA for samples from Sierra Leone and Nigeria may be due to the nature of the recombinant NP protein, from LASV AV, and the strain used to infect cells for IFA, the Bantou strain from Guinea. Using the same strain for the ELISA and the IFA might have led to more agreement and authors should discuss this point in their manuscript. They may provide information about similarities between the antigens. They may also discuss how their assay may gain specificity by using different NP proteins from different lineages.
Minor comments:
Line 25: dependent should be corrected to depend
Line 116: precise the nature of the NP antigen (strain, genbank ID, sequence)
Line 119: add a space for 24 hr
Line 137: give a genbank ID for the Bantou Strain
Lines 140 and 142: 37°C
Line 218: was not statistically significant
Author Response
We thank the reviewer 3 for this constructive review, and give our response below.
It is true that we had used the Bantou strain (lineage IV) to produce the IFA slides and test IgG in rodents from Guinea, Sierra Leone and Nigeria. As there is a cross-reaction between the arenaviruses, we believe that this does not influence the test result. In Guinea for example, the pygmy mice probably infected with a Kodoko virus (because one of them was Kodoko positive), thus very distant from the Lassa branch, reacted to the Bantou strain.
However, in line with your comment, we have introduced additional information concerning laboratory animals infected with 2 different strains, Bantou (lineage IV) and Kako (lineage VI). In animals infected with Kako, a distant lineage from the AV strain used in the ELISA, we did not observe any difference in specificity. This concept is now presented at the beginning of the discussion (lines 323-327). We also suggested to gain sensitivity (and not specificity which is already 100%) in repeating the negative ELISA with different dilutions of the blood/serum.
Other minor comments have been resolved, including the GenBank IDs of AV, Bantou and Kako strains.